# Principles of Treatment and Clinical-Evolutionary Peculiarities of Deep Cervical Spaces Suppurations—Clinical Study

**DOI:** 10.3390/life13020535

**Published:** 2023-02-15

**Authors:** Daniela Jicman (Stan), Nicolae Sârbu, Laura-Florentina Rebegea, Mihaela Crăescu, Elena Niculeț, Maria-Daniela Țuța, Aurel Nechita, Alin Codruț Nicolescu, Alin Laurențiu Tatu

**Affiliations:** 1Department of Otorhinolaryngology, “Sfântul Apostol Andrei” Emergency Clinical Hospital, 800578 Galați, Romania; 2Biomedical Doctoral School, Faculty of Medicine and Pharmacy, “Dunărea de Jos” University, 800010 Galați, Romania; 3Department of Radiology and Medical Imaging, “Sf. Ioan” Clinical Hospital for Children, 800487 Galați, Romania; 4Clinical Medical Department, Faculty of Medicine and Pharmacy, “Dunărea de Jos” University, 800008 Galați, Romania; 5Radiotherapy Department, “Sfântul Apostol Andrei” Emergency Clinical Hospital Galați, 800578 Galați, Romania; 6Research Center in the Field of Medical and Pharmaceutical Sciences, ReFORM-UDJ, 800010 Galați, Romania; 7Department of Morphology and Functional Sciences, Faculty of Medicine and Pharmacy, 800010 Galați, Romania; 8Department of Pediatrics, “Sf. Ioan” Clinical Hospital for Children, 800487 Galați, Romania; 9“Agrippa Ionescu” Emergency Clinical Hospital, 011773 Bucharest, Romania; 10Dermatology Department “Sfânta Cuvioasă Parascheva” Clinical Hospital of Infectious Diseases, 800179 Galați, Romania

**Keywords:** cervical deep spaces, treatment, management, particularities, complications

## Abstract

As medical-surgical emergencies, regardless of the causal agent, deep cervical space suppurations are not only a diagnostic challenge, but also a therapeutic one. In some cases, in spite of proper therapeutic measures, extremely severe complications can develop. A 5-year retrospective study (2016–2020) was conducted on a group of 107 patients suffering from cervical suppurations, being hospitalized and treated in the ENT Clinic of the “Sf. Apostol Andrei” Emergency County Hospital of Galați. This research is a clinical-statistical study based on the experience of this ENT clinic and was carried out based on the analysis of the patients’ medical records. Descriptive analysis’ statistical methods of the data series collected from the clinical observation sheets were used, with the patients’ informed consent for the processing of the aforementioned data, with the agreement of the Ethics Commission of the Emergency Clinical Hospital “Sf. Apostol Andrei” Galați and the College of Physicians Galați, România. The patients’ clinical and multidisciplinary treatment features included in the study group are presented. The results highlight the clinical particularities of deep cervical space suppurations treatment, including under COVID-19 impact, or with other comorbidities, having consequences on the case mix index increase or directly on the costs, admittance duration and the clinical status of the patient at discharge. The conclusions of the clinical study are based on the fulfillment of the research objectives in terms of treatment and symptomatology of deep cervical space suppurations and under the impact of comorbidities (global health crisis and pandemic, triggering of comorbidities due to health care access difficulty in the context of anti-COVID-19 government-implemented measures and the infection-rate that overburdened the medical system in the early period of the pandemic). Individualized treatment of deep cervical space suppurations is recommended to be approached multidisciplinary. Of particular importance is early diagnosis combined with prompt and correctly instituted multidisciplinary treatment. In this context, an appropriate medical measure that we recommend is patient health education, as it was observed in the clinical study: most times, patients address medical services with advanced disease, hence the generally unfavorable prognosis and outcome (about 25% of patients develop unfavorable prognosis and 4% die).

## 1. Introduction

Cervical deep space suppurations are defined as infections located in the fascial planes and spaces of the head and neck [1]. The anatomy of the cervical region is extremely complex, the anatomical structures belonging to this topographical area being enveloped by fasciae. The superficial and deep cervical fasciae (divided into 3 sub-fasciae) divide this region into interconnected compartments containing vital organs, blood vessels and nerves [1,2,3,4,5]. The superficial cervical fascia is made up of subcutaneous tissue and contains the platysma muscle. The deep cervical fascia is made up of 3 layers: superficial, middle and deep. These 3 subcomponents delimit cylindrical compartments that extend longitudinally from the base of the skull to the mediastinum. The superficial layer of the deep cervical fascia comprises a series of muscles and glands. The middle one, also called the pretracheal fascia, comprises the cervical organs such as the pharynx, esophagus, larynx, and includes the trachea, thyroid and parathyroid glands. The deep fossa, also called the prevertebral fossa, comprises the spinal column and its muscles [2,5,6,7,8,9]. Some authors classify the cervical spaces into supra-hyoid: parotid, masticatory, infratemporal, submandibular, parapharyngeal, peritonsillar and buccal spaces. The subhyoid space is also called the visceral space and there are spaces that extend, as mentioned above, from the base of the skull to the mediastinum, namely the prevertebral, retropharyngeal, dangerous and carotid spaces [9,10,11,12,13,14]. Suppurative processes frequently affect these compartments. The anatomic forms of these suppurations are cellulitis (which can be acute or chronic), abscess and phlegmon [14]. Cellulitis can be considered as the presuppurative stage which is characterized by vasodilatation phenomena with increased vascular permeability; it is tender to palpation but the area is not fluctuant. Abscess is a limited suppuration with the appearance of fluctuation. There are cellular changes in the overlying tegument. The patient is in an altered general condition, with the appearance of fever, chills, tachycardia and other toxico-septic phenomena. Phlegmon is an extensive, diffusely contoured suppuration characterized by the appearance of a massive, hard or fluctuant swelling or with crepitus on palpation. Vascular thrombosis and necrosis occur. The affected tissues acquire a siderated or infiltrated appearance of fetid purulent bloody secretions [11,15,16]. Deep cervical space suppurations are a heterogeneous group of infections with an incidence which continues to increase in spite of antibiotic availability. They are often underestimated or masked by other pathologies [17]. The incidence of these types of infections is estimated to be at approximately 10/100,000 population/year, with an increasing trend, especially in the pediatric population under 5 years of age, accounting for approximately 3, 400 hospitalizations annually in the United States [2,17]. According to some studies, they occur more frequently among the younger population (with an average age of 28 years) [18], but with an increasing trend in the elderly [19]. For example, according to a study conducted in the USA, the incidence of retropharyngeal abscess increased between 2003–2012 from 2.98 to 4.10/100,000 patients under 20 years of age, and that of peritonsillar abscess was 9.4/100,000 patients of the same age in 2009 with a peak registered in adolescents aged 13.6 years [17,20]. Other studies consider that the peritonsillar abscess is the most common cause of peripharyngeal space suppuration with a rather high incidence of about 30 cases/100,000 patients, occurring in both children and adults without much difference between race or sex [21]. The main starting points of these suppurative processes are odontogenic infections followed by infections in the ENT sphere which are neglected therapeutically or are incorrectly treated, presence of pharyngeal foreign bodies, external and internal trauma with pharyngoesophageal perforations, infectious diseases (scarlet fever, erysipelas, diphtheria etc.), the post-tonsillectomy state, salivary gland lithiasis, infected and abscessed benign and malignant tumors or ulcerated oral mucosal lesions [17,22,23,24]. Factors that aggravate the infectious processes of the deep cervical spaces are immunodepression, immunosuppression, neoplastic diseases which are under treatment, diabetes mellitus (D.M.), malnutrition, tuberculosis, pregnancy, liver cirrhosis, nephrotic syndrome, enteropathies, extreme age, alcohol abuse, smoking, psychotropic or narcotic substances [2,6,23,25]. Some studies state that these types of diseases frequently occur in rural populations, in patients with poor health education and unfavorable living conditions, or in populations that do not benefit from primary health infrastructures [26]. The etiology of lateral-cervical suppurations is described in the medical literature as being 85% polymicrobial, the most common being beta-hemolytic Streptococcus, Staphylococcus aureus and Pseudomonas aeruginosa. The germs involved are predominantly aerobic, but can also be anaerobic. Most times this microbial polymorphism results in a reciprocal protection of exogenous agents from the phagocytic process, and the intracellular “killing” process and antibiotics promote the necrotizing evolution of the disease, thus making it impossible to further isolate and identify the bacterial species involved [11,21]. The clinical manifestations of suppurations in the cervical region depend on the anatomical space affected, the presence or absence of abscess, the extent of the inflammatory process and local pressure, the type and virulence of the etiologic agent and the patient’s comorbidities [1,2,15]. The patient may have dyspnea and stridor, dysphonia, local or distant pain, febrile syndrome, dysphagia, odynophagia, otalgia, trismus, torticollis, facial rigidity, open rhinonitis. The most characteristic signs of a deep cervical suppuration are unilaterality, intense congestion, edema and local fluctuation with bulging of the wall of the affected anatomical structure, asymmetry of local elements, presence of crepitus or tender, woody, indurated skin tissue, sialorrhea, fetid halitosis, toxico-septic signs [6,21]. However, sometimes the diagnosis is difficult to establish, as some patients are asymptomatic [10], or they may present signs and symptoms characteristic of other common pathologies, or they may be masked by prior self-administration of analgesics, anti-inflammatory drugs or even antibiotics. For this reason these infections are often underdiagnosed or underestimated [17,27]. In order to properly diagnose deep cervical space suppurations, a complete clinical-biological workup, imaging techniques starting with cervical radiography, dental CT with attention to the lower molars 2, 3, cervical soft tissue ultrasonography, cervical CT, chest with contrast material, even MRI or angio-CT (in case of complications involving the neurovascular compartment or other spaces) [6,22,23,26,28,29] are needed. In some cases it may be necessary to perform histopathological examination to clarify the diagnosis [24]. Deep cervical space suppuration lends itself to extremely severe complications that can be life-threatening. The local anatomy with interconnected spaces and compartments allows the spread of microorganisms from one space to another, thus generating these complications. One of the most severe ones is considered to be mediastinitis [23]. The occurrence of this complication is explained by local conditions characterized by spaces limited by fasciae, with poor vascularization and poor presence of reticuloendothelial elements [21,30,31]. Dissemination of pathogens occurs by contiguity, lymphatic or haematogenous route. Mediastinitis is characterized by a severe infectious syndrome which is refractory to treatment, with the development of subcutaneous emphysema and interscapular pain, local crepitus, with congestion of the upper thoracic region, accompanied by dyspnea [21,23]. Other complications that may occur are: airway obstruction, especially laryngeal obstruction, aspiration pneumonia, cranial nerve paralysis (especially IX-XII, the cervical sympathetic nerve) with the appearance of nerve syndromes. Thrombosis or major vascular aneurysms (of the carotid artery or jugular vein) and serious infectious phenomena may also occur: necrotizing fasciitis, disseminated intravascular coagulation empyema, septic shock, pulmonary embolism, death [1,6,17,23]. Of major importance among the risk factors influencing the occurrence and evolution of deep cervical space suppuration is alcohol consumption, which also influences the duration of hospitalization [32].

## 2. Materials and Methods

This paper presents a retrospective study covering a period of 5 years, between 2016 and 2020 respectively. In this study we included a group of 107 patients. Patients over 18 years of age were included in the study, giving their consent to participate by signing the informed consent in the observation sheets. Patients diagnosed with cervical suppurations, regardless of etiology and gender, were included. Patients under 18 years of age, pregnant women, patients with infected head and neck tumors and those suffering from superficial skin abscesses were excluded. Iatrogenic or posttraumatic infections of the neck and patients whose data were incomplete were also excluded. All study participants were examined on the basis of history, clinical examination and investigations.

The following clinical characteristics were analyzed and compared: age, sex, symptoms, comorbidities, lifestyle habits (smoking, alcohol, drugs), etiology of suppurative processes, affected area, bacteriological and blood tests, specific imaging. Treatment, type of administered antibiotic, symptomatic and surgical treatment were also analyzed factors. Complications, need for tracheostomy and hospitalization times were noted.

Statistical methods were used for the descriptive analysis of the data series collected from clinical observation records. All parts of the present study were conducted in accordance with the guidelines of the Declaration of Helsinki and approved by the ethics committee of the Emergency Clinical County Hospital “Sf. Apostol Andrei “Galați with no: 16141/24.06.2022, as well as by the ethics committee of the College of Physicians, Galați, România, no: 699/15.06.2022. This research is a clinical-statistical study based on the experience of the ENT clinic and carried out based on the analysis of patient observation sheets.

Concerning statistics, we used XLSTAT software for correlations between the variables analyzed. As function, we used Principal Component Analysis and Pearson coefficient calculation. Values of *p* < 0.05 were considered as statistically significant.

## 3. Results

In this study we included a group of 107 patients who were hospitalized and treated in the ENT Clinic of the Emergency Clinical County Hospital “Sf. Apostol Andrei “Galați, România (see Table 1).

Patient comorbidities in the studied group were present in 89.71% of cases, most of them having cardiac pathology 35.51%, pulmonary pathology 14.02%, being present also cases of DM and dental infections. Obesity was recorded in 7.47% of patients, but we found also patients without any pathology 10.28%. Also noteworthy is the high percentage of patients with a previous history of ENT suppurations, 43.93% and the high number of patients who smoke 56.07% (see Table 2 and Table 3).

The main starting points of cervical suppurations in our study are bacterial tonsillitis in a percentage of 25.23%, followed by odontogenic causes 23.36%, lymphadenitis 14.01% and also superinfected cysts or complicated ENT conditions, such as otitis media or upper respiratory tract infections, in a proportion of 2.80% and, respectively, 3.74% of all patients; there were also unidentified causes in 10.28% patients (see Table 4).

The clinical characteristics of the studied group is complex, ranging from dysphagia, with the highest percentage of 95.32%, throat edema 90.65%, sore throat 87.85%, to fever and otalgia 9.34% or airway obstruction in 3.73% of patients (see Table 5).

The main spaces involved in the studied patients’ suppurations were: the submandibular space which had the highest rates, with a percentage of 25.23%, being more common in the age groups over 51 years (13 cases), followed by the buccal plane space, sublingual one with a percentage of 23. 36%, peritonsillar space 18.69% and patients with multispatial involvement in 15.88% of cases, the lowest rates being recorded for parapharyngeal or retropharyngeal spaces, in a percentage of 13.08% and 3.73%, respectively.

In addition, 52 patients had the suppurative process located on the right side, i.e., 48.59% and, 47 patients on the left side 43.92%; in 8 patients there was bilateral involvement 7.47% (see Table 6).

The etiological agents responsible for the occurrence of cervical suppurations in the studied group were, in order of frequency: Staphylococcus aureus 28.03%, followed by Streptococcus Group G 15.88%; Gram negative bacteria were also found to be involved: Klebsiella, Pseudomonas aeruginosa or anaerobic species such as Fusobacterium necrophorum; negative cultures were also recorded in 22.43% of cases (see Table 7).

All patients had inflammatory markers, bacterial cultures and cervical X-rays, ultrasound or CT scans taken and compared.

The most relevant values for inflammatory markers are shown in Table 8, considering that we compared the data to the normal values used in our hospital’s laboratory: Erythrocyte sedimentation rate (ESR) with normal values between 2–15 mm/h, Fibrinogen between 200–400 mg/dL, C-reactive protein (CRP) between 0–5 mg/L, Procalcitonin (PCT) between 0–0, 10 ng/mL and Leukocyte between 4–9 × 10^3^ (see Table 8).

Treatment was multidisciplinaryin many cases, taking into account the starting point of the suppuration. Of the studied group, 10.28% of the patients had suppuration that only recovered under antibiotic and symptomatic treatment, in 74.76% cases surgical drainage was needed, even multidisciplinary treatment. Spontaneous fistula formation was also recorded in 11.21% of patients. In the studied group, 4 deaths were also recorded (see Table 9).

The antibiotic regimen consisted of double or triple therapy (see Table 10), but in the presence of other comorbidities and complications, multiple therapies were used. The most commonly used antibiotics administered in combination were: cefortum, gentamicin and metronidazole with remarkable results. Sulcefate, oxacillin, imipenem, clindamycin, piperacycline/tozabactam were also used. The average in-hospital admittance was 10 days, maximum from 6–8, to even 10–14 weeks in patients with complications and with SARS-CoV-2 superinfections. We also emphasize the hospitalization costs related to our study which were extremely high, especially in cases with complications, costs that are closely correlated with the duration of the hospital stay. In some cases, the costs ranged from 4400 euro to 12,409 euro or even 15,437 euro. In addition, during the treatments, maximum attention was paid to surgical wound cleaning and cavity washing with diluted antiseptic solutions or antibiotic ones, especially for patients with submandibular drainage tubes or other sites. Antibiotic treatment was associated with analgesic, anti-inflammatory, hemostatic, cardiotonic treatment, vitamin therapy, sedative treatment in immobilized patients suffering from complications and in those admitted to the intensive care unit (n = 20, i.e., 18.69%).

40.18% of the patients studied developed complications that required a multidisciplinary effort: mediastinitis, necrotizing fasciitis, aspiration pneumonia, septic shock and airway obstruction, and 4 patients required emergency tracheostomy (please study the examples in suggestive examples of patients affected by deep cervical space suppuration). Of all 107 patients studied 23 had SARS-CoV-2 infection representing a percentage of 21.95%. Patients with complications and CoV-2 superinfection were isolated in special intensive care units (see Table 11).

Correlations were made between the comorbidities of the patients in the studied group and the occurrence of complications. Thus, statistically significant correlations were found between presence of diabetes and the following complications: mediastinitis (*p* = 0. 032), necrotizing fasciitis (*p* = 0.001), septic shock (*p* = 0.006) and intensive care unit admission (*p* < 0.0001). Pearson coefficients for these three correlations are 0.208, 0.325, 0.264 and 0.372 respectively. Other correlations found were between presence of liver and mediastinal disease (*p* = 0.037), and admission to the intensive care unit (*p* = 0.042) (see Table 12).

Correlations were also made between the patients’ comorbidities and number of deaths (4/107 patients). Deaths were more frequent in patients with diabetes and lung disease. However, a statistically significant correlation, *p* < 0.0001, was only found in patients with diabetes as an associated disease. The Pearson correlation coefficient was 0.377 (see Table 13).

Statistically significant correlations were also found between the presence of Streptococcus pyogenes or Fusobacterium necrophorum, respectively, with the association of more than 3 antibiotics (*p* = 0.007 and *p* < 0.0001 respectively). Pearson correlation coefficient is 0.258 and 0.502 respectively (see Table 14).

Regarding the correlations between the therapeutic regimen and the location of suppuration, we found correlations without statistical significance between the suppuration location in the submandibular space and spontaneous fistula, and between the suppuration location in the parapharyngeal space (*p* = 0.13), and healing only under drug treatment (*p* = 0.09) (see Table 15).

Suggestive examples of patients affected by deep cervical space suppuration.


**Case 1.**


A 79-year-old patient presenting to our clinic with left hemifacial oedema, extended in the submandibular, laterocervical to left supraclavicular fossa, anterior cervical, with oedema at the left hemifacial level extended to the base of the tongue, aryepiglottic fold and left pyriform sinus. Multidisciplinary intervention with mounting of drainage tubes. Despite the institution of an adequate management the evolution was unfavorable with the onset of septic shock, later death (see Figure 1).


**Case 2.**


A 45-year-old patient, smoker, presented in the emergency department suffering from severe inspiratory dyspnea, marked trismus, dysphagia, odynophagia, dysphonia, fever and cough. On inspection and palpation there was bilateral submandibular painful filling, without clinically palpable adenopathy. Buccopharyngoscopy visualizes congestion of the right lateral pharyngeal wall, posterior to the right tonsillar box, extending to the hypopharynx. The right palatine tonsil appears pushed towards the midline and hypertrophied comparable to the left, with luteal edema. Indirect laryngoscopy shows a heavily congested laryngeal crown, inflammatory edema of the epiglottis and aryepiglottic folds; glottis cannot be fully visualized, possible epiglottic abscess—an abscessed spot is present on the laryngeal side of the epiglottis. Subsequently, 2 days later, epiglottic edema is visualized with an area covered by purulent ulcero-necrotic membranes through which fetid-purulent secretions are removed. The epiglottis phlegmon is surgically opened and the newly formed collection of the posterior pharyngeal wall (posterior to the left posterior tonsillar pillar) is removed, eliminating purulent secretions. Subsequently there is a spontaneous abscess from the right lateral hypopharyngeal wall with spontaneous daily elimination of purulent secretions. The patient suffers from diffuse thoracic pain. He was transferred from the ENT clinic to the thoracic surgery department where emergency surgery was performed and the collection was drained with pleuro-visceral decortication, with removal of fibrinous, purulent tracts from the anterior and posterior mediastinum with lavage and installation of a drainage tube. Surgical reintervention and right iterative thoracotomy, pleuropericardial window, then pleuro-visceral decortication with right thoracotomy, repeated ENT-cervicotomy surgery, with rigorous grooming and local dressing were performed (see Figure 2).

The evolution was insiduous but favourable. He was discharged in good general condition, with an expanded lung on the control X-ray.

CT examination find the following changes (see Figure 3).


**Case 3.**


54 years old patient, smoker, alcohol consumer, presents with dysphagia, odynophagia, left submandibular pain, reflex otalgia and marked trismus. Multidisciplinary oral and maxillofacial surgery consultation: left submandibular adenophlegmon without a dental cause (see Figure 4).

Locally, left submandibular swelling is observed, extended in the submenton, imprecisely delimited. After partial remission of the trismus, clinical examination showed tongue and left posterior tonsillar pillar oedema, swollen, congested epiglottis, mobile right vocal cord, diminished but sufficient glottis; left vocal cord cannot be visualized; left lateral hypopharyngeal wall oedema and left valecula, left pyriform sinus with salivary stasis at this level. Under triple antibiotic and symptomatic therapy for 2 weeks, persistence of left palatine tonsil congestion is visualized, which is slightly pushed towards the midline, the edema of the luteum and tonsillar pillars being remitted, without signs of collection in the oropharynx, without celsius signs present but with persistence of submandibular and submental skin infiltration. Under treatment the evolution is favorable with remission of symptoms and local inflammatory phenomena (laryngeal and cervical). Subsequently, the tracheal stoma is closed with resumption of physiological breathing. The patient is discharged clinically cured. The CT examination of this case shows the changes found in Figure 5.


**Case 4.**


A 26-year-old patient presents in the emergency department with mixed dysphagia, odynophagia, marked trismus, reflex otalgia, mucopurulent secretions in the oral cavity, which are difficultly eliminated by the patient, and right submandibular pain. The patient also presented with hyperemic right palatine tonsil with bulging of the posterior tonsillar pillar, bulging of the lateral pharyngeal wall with narrowing of the bucco-pharyngeal isthmus. Surgical intervention (maxillo-oral surgery) and submandibular incision with multiple drainages (9 drainage tubes, Figure 6a). The evolution was unfavorable with bilateral loco-cervical anterior extended cervical phlegmon on vascular tract, anterior and posterior mediastinum up to half of the sternal manubrium. Surgical intervention (ENT surgery) and 3 bilateral horizontal stepped cervicotomies are performed, with evacuation of purulent collections, sterile tables are introduced at the level of the drainage cavities. Afterwards a suprasternal cervicotomy was performed with control of the mediastinal lobe up to half of the sternal cuff, with evacuation of the left preclavicular abscess, cervico-mediastinal and preclavicular drainage, suturing of the suprasternal cervicotomy (Figure 6b) (thoracic surgery). The unfavorable evolution continued, with continued left upper antero-posterior extended mediastinum and periaortic, left pleurisy. Surgical intervention was performed with extraction of dental foci 1.6, 3.7, 3.8, and removal of submandibular drainage tubes (Figure 6c). Thoracic surgical approach included left thoracotomy in the 3rd intercostal space with evacuation of hyperchromic serous fluid with fibrin clots, upper mediastinal and periaortic pleurectomy at the level of the aortic cross and in the upper ¾ of the descending aorta. Bilateral anterior cervical drainage tubes are inserted in the course of the vascular sheaths (2 tubes), along with anterior and posterior cervical-mediastinal pleura (2 tubes), anterior and posterior pleuro-mediastinum (2 tubes). The evolution is slowly favorable, complicated by infection with SARS-CoV-2. The patient was discharged SARS-CoV-2 negative, with bilateral expanded lung, no intrapleural fluid, normal mediastinal silhouette. Patient at final evaluation (Figure 6d,e).

CT imaging finds the changes seen in Figure 7.

## 4. Discussion

The literature states that any infection located in the skin or deep soft tissues, having a rapid evolution and negative prognosis, should benefit from aggressive therapeutic management [33,34]. Deep cervical space suppurations are difficult to diagnose in their early stages [6], therefore each patient is a particular case [30], sometimes requiring atypical management [23].

In a study carried out in India, on a sample of 45 patients, 67% of the rural male population (69%) was diagnosed with deep cervical space suppurations, and 64.11% of the rural population was diagnosed with deep cervical space suppurations; odontogenic infection was the primary cause of this pathological entity [26]. On the other hand, Velhonoja et al. confirms in his retrospective study on 277 patients that the odontogenic etiology, i.e., 44.8% was most frequently found with a clear male predominance of 179, (64.6%) in the studied group, because men have poorer dental conditions, as compared to women (i.e., in 76 (75.2%) of the studied patients) [14].

However, in our study the first cause was fund to be peritonsillar infection with a number of 27 patients (25.23%), odontogenic infection being found in 23.36% of patients, quite close in ratios, a result similar to the study of Brito et al. which identified a percentage of 31.68% and odontogenic infections with 23.7% of 101 patients in the study [17]. Regarding the frequency rate of the anatomical regions involved in the suppurative processes, the literature describes a predominance in submandibular space involvement, with a number of 317 patients, respectively 54.1%, in a retrospective study by Mejzlik, J et al. from a group of 586 patients, followed by the masseteric space with 131, patients, i.e., 22.4% [35], as demonstrated in the study by Das, R et al., in which the submandibular space was affected in 66.6% of cases, a situation found in 30 out of 45 patients [26]. The same situation resulted from our study, where out of 107 patients, 27, i.e., 25.23% were diagnosed with suppuration at this level. However, there are studies that note that adults, due to associated comorbidities, are often prone to multispatial infections [17].

Among the patients’ comorbidities suffering from cervical suppurations, studies list D.M., high blood pressure, pulmonary disease, cardiac disease [1,17,26]. It is emphasized that in patients diagnosed with suppurative processes, D.M., alcohol use, including cirrhosis of the liver predispose patients to extremely adverse complications such as necrotizing fasciitis [33]. Moreover, it stated that glycemic level influences clinical outcomes, being associated with an increased incidence of complications or longer hospitalization periods [36]. There is even research showing that D.M. increases the risk of developing mediastinitis almost 5 times more than non-diabetic patients [25]. A meta-analysis by Hidaka et al. confirms that diabetic patients diagnosed with deep throat suppuration have multispatial localization which is frequently associated with Klebsiella pneumoniae or anaerobic germs [37]. In our study, out of 107 patients, 25, respectively 23.40% suffered from diabetes mellitus, and out of this small group, the highest frequency was found among those of 51–60 years of age, respectively 25.00%. Among the patients with D.M., there were 10 infections with SARS-CoV-2, i.e., 9.35%, 2.80% i.e., 3 patients out of 25 died, 1.87% had necrotizing fasciitis and aspiration pneumonia and 0.93% had complications such as mediastinitis and airway obstruction.

In order to avoid these complications, early diagnosis is necessary, and to obtain a reliable diagnosis quality sampling is required, making use of specialized and invasive techniques that are not always available [25,38].

According to current studies, therapeutic success in deep cervical space suppurations is based on prompt and correct management [1,6,25].

Mandatory treatment principles include a broad-spectrum, single, double or triple antibiotic regimen in combination with surgical drainage (which should only be performed if purulent collection is found, as premature incision may influence local defense system and may hasten the local or distant spread of infection) [2,6,23]. 

Effective management must take into account the patient’s age, infectious process etiology, origin and course of spread, specific antibiotic sensitivity of the causative agent, and comorbidities present. For example, the presence of peripheral vascular disease caused by D.M. predisposes patients to anaerobic infection [39]. The choice of antibiotic regimen requires intense medical judgement, considering the possibility of etiological agents’ resistance. Treatment should be administered correctly in maximum doses with synergistic effects on bacterial populations and to optimize tissue penetration [6,10,23]. The duration of antibiotic regimens, according to medical studies, generally lasts between 2 and 3 weeks, with a visible effect on biological constants and local appearance. If complications involving vital anatomical structures are present or the etiological agents are anaerobic in nature or are multiresistant, the duration of treatment should be prolonged and doses and drug combinations adjusted until the desired efficacy is achieved. Broad-spectrum molecules with maximum bioavailability and normal absorption are preferred, so that first-line therapy must be reviewed after 48 h and adjusted according to the antibiogram criteria [1,6,14,17,23,26,35]. 

In our study, broad-spectrum antibiotics were used as double therapy in 25 patients, i.e., 23.36%. In 62 patients, i.e., 57.94%, representing the majority of the patients included in our study, antibiotics were administered as triple therapy, and in 10 patients, i.e., 9 patients, multiple therapy was administered. Among the classes of antibiotics used successfully in our study are found: beta-lactams, higher generation cephalosporins, aminoglycosides, carbapenems, lincosamides, glycopeptides, metronidazole and antiviral agents in case of complications with SARS-CoV-2. One of the fundamental measures in the treatment of suppurations, according to several studies, is incision and drainage of these collections, but, sometimes, in spite of highly advanced imaging investigation techniques, a preoperative diagnosis can often be missed [40].

In patients with unilocular abscesses, minimally invasive techniques are successfully used to avoid aesthetic complications. Sometimes, the needle aspiration drainage of abscesses is also effective and studies show this therapeutic method to be beneficial in reducing morbidity of surgery and trauma, reducing healing time, shortening the hospital stay, reducing risk of contamination of healthy perilesional tissue.

Drainage techniques are varied and are adapted to each patient’s case, and are also depended on the location of the purulent collection. Sometimes, transcervical drains, cervicotomies or other techniques are necessary in concordance with the extent of the infectious processes [1,6,41].

We believe that an essential principle is multidisciplinary management—a multimodal treatment that should be instituted immediately after diagnosis [6,23,42].

Moreover, it has been found that the timing and modality of surgical drainage is a determining factor for the therapeutic success, and in some cases deep debridement with reconstruction requiring multidisciplinary effort is necessary [23,25,33,43]. In our study, 80 patients, i.e., 74.76% required surgical drainage; in 11 patients (10.28%) the suppurative process was remitted under treatment, and in 11.21% of patients the suppurations had spontaneously created a fistula.

One of the most controversial principles in the management of suppurative processes is the administration of anti-inflammatory drugs. Some authors consider that administering them should only be done in the precollection phase [2,23,28].

In a retrospective observational descriptive study of 365 patients, Boscolo-Rizzo et al. also emphasized the importance of airway control as a fundamental part in the management of suppurative processes [41]. Thus, in our study 4 patients required tracheostomy.

Complex management includes among many other measures, also the management of: pain, identification and removal of the causative focus—in case of suppurations of odontogenic origin [14], multidisciplinary management of complications—sometimes patients require special care in intensive care units, alleviation of nutritional deficiency and strengthening of the compromised immune system [6,12,23,44]. In our study patients were administered antibiotic therapy, underwent surgery and were given haemostatic agents, vitamin therapy, and cardiac support was insured. Out of 107 patients 18.69% required special care in the intensive care unit.

A relatively new element of symptomatic treatment is, according to the literature, the hyperbaric oxygen therapy with antibacterial purpose, used for remission of edema and stimulation of tissue neoformation processes, thus reducing subsequent debridement surgery [21,33,35,45]. In our study this technique was not used.

Studies in this field place emphasis on complications and their management. Early surgical intervention together with correctly instituted treatment is precisely aimed at preventing the occurrence of complications associated with deep cervical space suppuration [6,17,26,35]. 

In our study, out of 107 patients studied, 23 also had SARS-CoV-2 infection, with a percentage of 21.95%.

Infection with the new SARS-CoV-2 had a particular impact on patients with cervical suppurations, especially in those patients already suffering from comorbidities. This was highlighted by the fact that the suppurative process was accelerated in some cases, lung damage prolonged hospitalization, and increased the related costs. In these patients the evolution was generally slower, sometimes favorable, but deaths were recorded.

Current studies show that in patients infected with severe SARS-CoV-2, there is a biomarker that indicates the severity and prognosis of the disease; such a biomarker is Interleukin 6, whose level is very high especially in patients treated in the intensive care unit.

Interleukin 6 is also a suggestive marker being involved in the infection-associated inflammatory response; it is a “stress cytokine” whose concentration increases in patients affected by suppurations or in those undergoing surgery; it is a marker that can guide therapeutic decisions [46,47]. 

There is always a possibility of developing several primary elements in the same tissue—a fact which shouldn’t be ignored [48]. 

Deep cervical space suppurations have an unpredictable course, often with a poor prognosis; therefore, special attention should be paid to prevention methods. These measures should be implemented especially among rural or low-income populations with low access to health care and poor health education. For this reason, more attention should be paid to physical health, especially dental hygiene, with the early management of dental caries, even from an early age [2,10,12,35,49]. At the same time, psychological support and continuous monitoring of patients with suppurative pathologies should not be neglected [25]. 

## 5. Conclusions

Currently, the management of deep cervical space suppurations is a diagnostic and therapeutic challenge, often being underestimated. These processes, in spite of an adequately instituted treatment, can have unpredictable outcomes and can develop extremely severe complications. Sometimes, suppurations caused by agents with antibiotic multiresistance can develop, hindering the medical act and influencing the disease prognosis. Given the complex anatomy of the area, sometimes risky surgical measures are required, multidisciplinary collaboration being needed. In our experience, the cornerstone for managing these pathologies rests in their early diagnosis, broad-spectrum antibiotic therapy, in combined regimens instituted immediately upon health care services addressing, accompanied by prompt drainage of the collection, identification and elimination of causative factors and resolution of the favoring ones, supportive medication administered depending on the patient. Comorbidities directly influence the evolution of the disease and may favor the development of complications. Even if the patient is diagnosed with the use of modern technologies and receives adequate treatment, even with a succesful surgical approach, serious complications (even death), may occur. Due to the aforementioned unfavorable disease course, effective multidisciplinary management protocols need to be developed and implemented, thus reducing morbidity and hospital costs. Preventive measures are needed for these types of infections, especially those with odontogenic starting point; this should be aided by the effective population screening methods, leading patients to a more careful oral cavity health monitoring. One of the immediate measures needed to be implemented through multidisciplinary effort is patient health education. They often address the specialist suffering from advanced forms of disease, hence compromising prognosis and outcome. Deep cervical space suppurations are a medical and surgical emergency and still represent a major diagnostic and treatment problem.

## Figures and Tables

**Figure 1 life-13-00535-f001:**
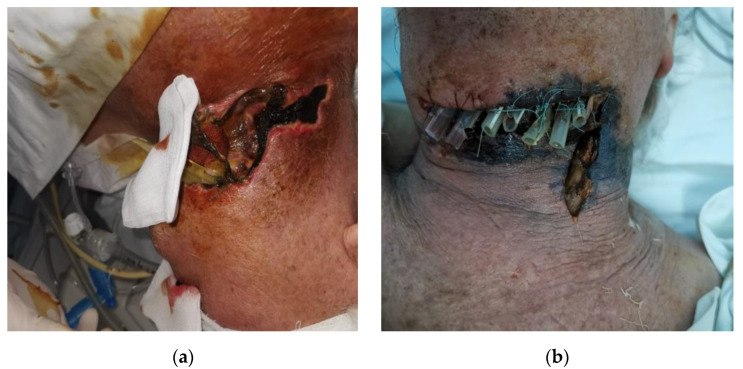
(**a**,**b**) phlegmon of buccal plane extended at cervical level, necrotizing fasciitis observed, probably odontogen starting point.

**Figure 2 life-13-00535-f002:**
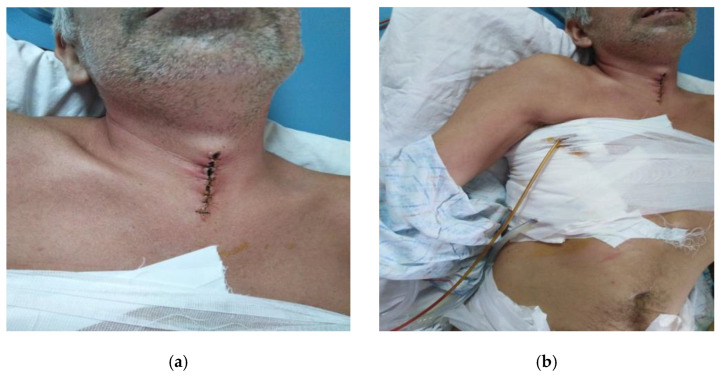
(**a**,**b**) Inflammatory process at epiglottic level, superinfected, with right peritonsillar starting point complicated with necrotizing descending mediastinum and right pleural empyema.

**Figure 3 life-13-00535-f003:**
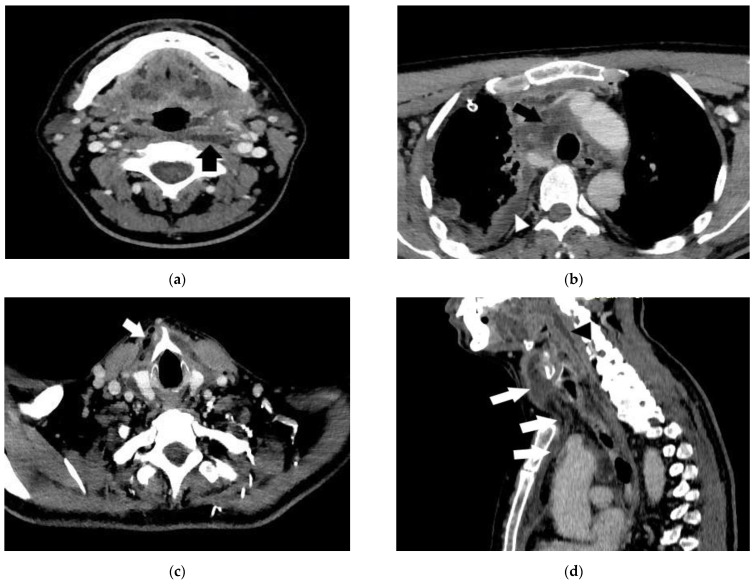
(**a**–**d**): Necrotizing descending mediastinitis: (**a**) Axial CT—section at lower oropharynx (suprahyodian) shows multiple fluid collections, one of which is located in the retropharyngeal space (black arrow). (**b**) Axial CT—section at mediastinal level shows fluid collections in the pericardial fat and right pleural cavity (black arrow). (**c**) Axial CT—section at the infraglottic floor after a few days shows gas bubbles in the fat in the anterior cervical space (white arrow). (**d**) Sagittal plane reconstruction—shows continuity solution between collections from cervical to mediastinal level (white arrows) and collection at retropharyngeal space level.

**Figure 4 life-13-00535-f004:**
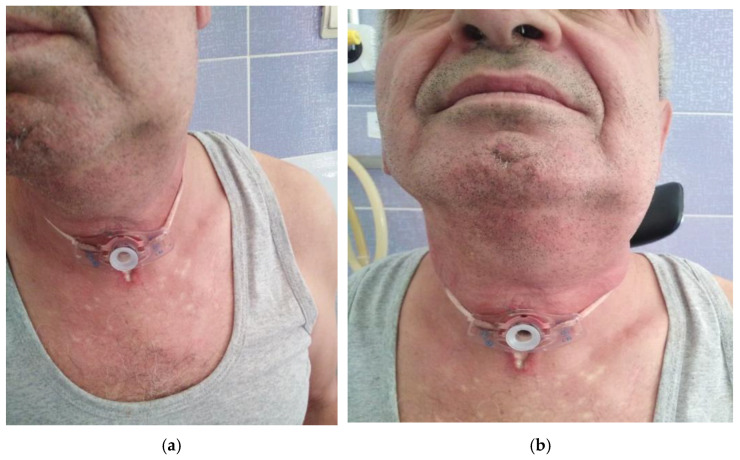
(**a**,**b**): Left peritonsillar phlegmon complicated with left parapharyngeal abscess, necessity tracheotomy.

**Figure 5 life-13-00535-f005:**
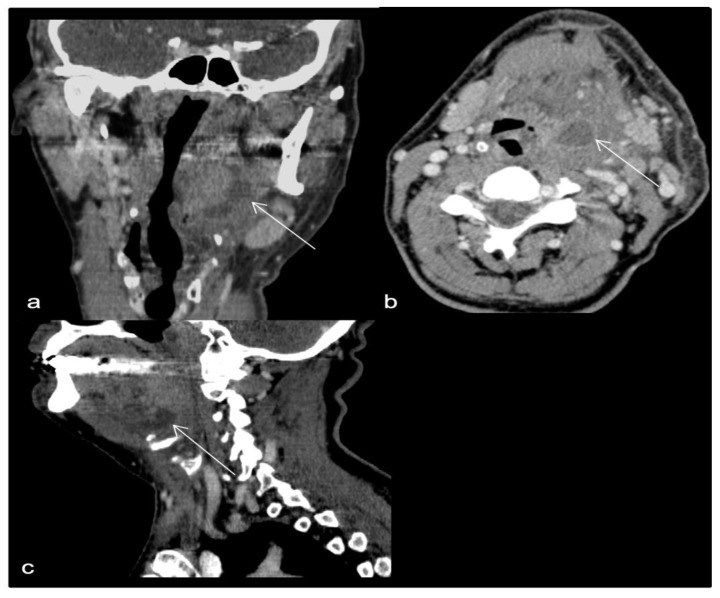
(**a**–**c**): Contrast enhanced CT scan of the neck with coronal (**a**), axial (**b**) and sagittal (**c**) reformats CT of the neck showing an expansile ill-defined soft tissue mass (arrows) with peripheral contrast enhancement and central hypodense non-enhancing areas indicating an abscess, measuring 44 × 40 × 60 mm (antero-posterior × transverse × vertical diameters), with mass effect and oropharyngeal displacement. The lesion is extending from the left parapharyngeal space to the superior level of the paralaryngeal space, and has mass effect over the epiglottis, the left vallecula and the piriform recess.

**Figure 6 life-13-00535-f006:**
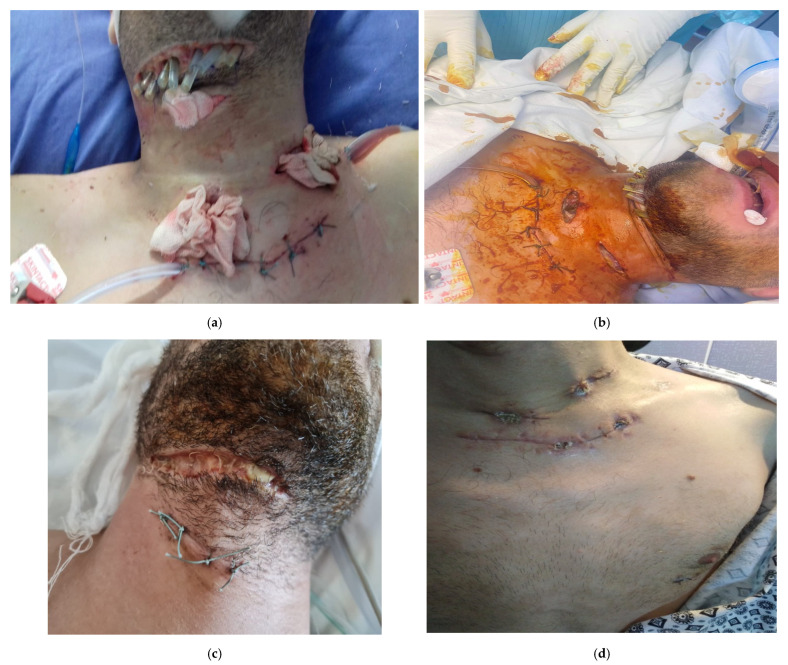
(**a**–**e**) Suppurated apical periodontitis 3.7.generated in the buccal phlegmon planseum infected with Pseudomonas aeruginosa and Fusobacterium necrophorum, bilateral anterior cervical extension complicated with left preclavicular abscess, mediastinitis, left pleurisy with fibrin clot, septic syndrome. SARS-CoV-2 infection with visible lung lesions on CT.

**Figure 7 life-13-00535-f007:**
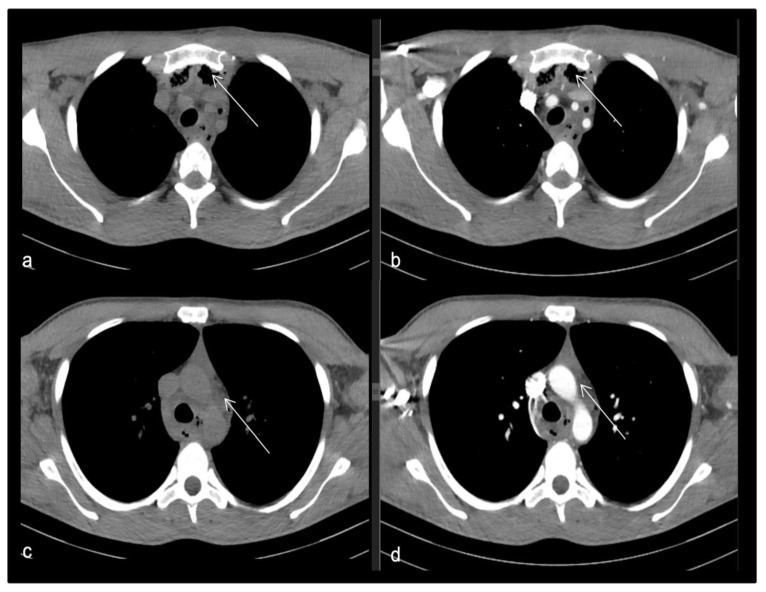
(**a**–**d**): Thoracic CT scan with axial reformats CT images of a 26-year-old male (**a**,**c**—non-enhanced CT; **b**,**d**—arterial phase contrast-enhanced CT) demonstrating mediastinal gas collection (arrows in images **a** and **b**) and fat stranding at the level of superior and anterior mediastinum (arrows in images **c** and **d**), indicating mediastinitis.

**Table 1 life-13-00535-t001:** Demographic characteristics of patients.

Age Groups	No. of Patients (%) N = 107
18–20 years	3 (2.8)
21–30 years	15 (14.01)
31–40 years	17 (15.88)
41–50 years	13 (12.14)
51–60 years	28 (26.17)
61–70 years	18 (16.82)
71–80 years	13 (12.15)
>80 years	0
Sex	
Female	48 (44.86)
Male	59 (55.14)
Environment	
Rural	74 (69.16)
Urban	33 (30.84)

**Table 2 life-13-00535-t002:** Factors that influenced the evolution and therapeutic conduct of the group of patients studied.

Comorbidities	No. of Patients (%) N = 107
History of suppurations	47 (43.93)
Cardiovascular diseases	38 (35.51)
Pulmonary diseases	15 (14.02)
Diabetes mellitus	25 (23.36)
Liver diseases	12 (11.21)
Hematological diseases	1 (0.93)
Dental infections	15 (14.00)
Obesity	8 (7.47)
Any pathology	11 (10.28)

**Table 3 life-13-00535-t003:** Other factors influencing the course and development of complications.

Other Factors Influencing Evolution and Complications	No. of Patients (%) N = 107
Smoking	60 (56.07)
Alcohol	14 (13.08)

**Table 4 life-13-00535-t004:** The main starting points of the suppurations in the studied group of patients.

Starting Points of Suppurations	No. of Patients (%) N = 107
Bacterial tonsillitis	27 (25.23)
Odontogenic cause	25 (23.36)
Lymphadenitis	15 (14.01)
Tonsillo-pharyngitis	8 (7.47)
Superinfected thyroid cyst	6 (5.60)
Over infected sebaceous cyst	5 (4.67)
Complicated otitis media	3 (2.80)
Upper respiratory tract infections	4 (3.74)
Sub-maxillitis	3 (2.80)
Unidentified causes	11 (10.28)

**Table 5 life-13-00535-t005:** Signs and symptoms of patients in the study group.

Signs and Symptoms	No. of Patients (%) N = 107
Neck pain	94 (87.85)
Neck swelling	97 (90.65)
Dysphagia	102 (95.32)
Odinophagia	93 (86.91)
Toothache	15 (14.01)
Trismus	89 (83.17)
Limited neck mobility	73 (68.22)
Dyspnea	9 (8.41)
Fever	19 (17.75)
Limphadenopathy	32 (29.90)
Ear pain	10 (9.34)
Airway obstruction	4 (3.73)

**Table 6 life-13-00535-t006:** Location of suppurative processes and spaces involved.

Location	No. of Patients (%) N = 107
Right side	52 (48.59)
Left side	47 (43.92)
Bilateral	8 (7.47)
Peritonsillar space	20 (18.69)
Submandibular space	27 (25.23)
Para-pharyngeal space	14 (13.08)
Buccal floor, sublingual space	25 (23.36)
Retropharyngeal space	4 (3.73)
Multispace	17 (15.88)

**Table 7 life-13-00535-t007:** Etiological microorganisms of deep suppurations from the studied group.

Cultured Bacteria	No. of Patients (%) N = 107
Staphylococcus aureus	30 (28.03)
Streptococcus Group G	17 (15.88)
Streptococcus viridans	14 (13.08)
Klebsiella	9 (8.41)
Pseudomonas aeruginosa	6 (5.60)
Streptococcus pyogenes	4 (3.73)
Fusobacterium necrophorum	3 (2.80)
Negative culture	24 (22.43)

**Table 8 life-13-00535-t008:** The highest biological values recorded.

	2016	2017	2018	2019	2020
Erythrocyte sedimentation rate (ESR) (mm/h)	100	108	92	105	112
Fibrinogen (mg/dL)	796	850	959, 4	642	1037, 6
C-reactive protein (CRP) (mg/L)	28, 1	32, 5	15, 2	92, 1	1037, 6
Procalcitonin (PCT) (ng/mL)	1, 19	1, 20	1, 15	2, 18	2, 14
Leukocyte (×10^3^)	16, 69	20, 31	19, 89	37, 45	25, 66

**Table 9 life-13-00535-t009:** Evolution of cases under treatment.

Case Management	No. of Patients (%) N = 107
Remedy under treatment	11 (10.28)
Drainage with surgery	80 (74.76)
Spontaneous fistula	12 (11.21)
Deceased	4 (3.73)

**Table 10 life-13-00535-t010:** The therapeutic regimen used.

The Treatment Regime Applied	No. of Patients (%) N = 107
Dual therapy	25 (23.36)
Triple therapy	62 (57.94)
More than 3 antibiotics	10 (9.34)

**Table 11 life-13-00535-t011:** Complications occurring in the group of patients studied.

Complications	No. of Patients (%) N = 107
Upper airway obstruction	4 (3.74)
Mediastinitis	5 (4.67)
Necrotizing fasciitis	3 (2.80)
Aspiration pneumonia	6 (5.60)
Septic shock	2 (1.87)
Death	4 (3.74)
SARS-CoV-2	23 (21.95)

**Table 12 life-13-00535-t012:** Correlation matrix between associated comorbidities and complications.

Correlation Matrix (Pearson):								
Variables	Upper Airway Obstruction	Mediastinitis	Necrotizing Fasciitis	Septic Shock	Aspiration Pneumonia	SARS-CoV-2	Intensive Care	Deceased
Cardiovascular disease	0.163	−0.072	0.111	−0.102	0.074	−0.088	−0.072	−0.043
Pulmonary diseases	0.062	−0.089	0.094	0.143	−0.098	0.061	0.072	0.062
Diabetes mellitus	0.017	0.208	0.325	0.264	0.070	0.072	0.372	0.377
Liver diseases	0.086	0.202	−0.060	−0.049	0.171	−0.108	0.197	0.086
Hematological diseases	−0.019	−0.022	−0.016	−0.013	−0.024	−0.049	−0.048	−0.019
Dental infections	−0.083	0.156	0.088	0.136	0.012	0.046	0.189	0.056
Obesity	−0.056	−0.063	−0.048	−0.039	−0.069	0.119	−0.051	−0.056
Any pathology	−0.067	−0.075	−0.057	−0.047	−0.083	−0.020	−0.167	−0.067

**Table 13 life-13-00535-t013:** Correlation matrix between comorbidities and deaths.

Correlation Matrix (Pearson):									
Variables	Cardiovascular Disease	Pulmonary Diseases	Diabetes Mellitus	Liver Diseases	Hematological Diseases	Dental Infections	Obesity	Any Pathology	Deceased
Cardiovascular disease	1	−0.243	−0.198	−0.140	−0.072	−0.202	−0.062	−0.251	−0.043
Pulmonary diseases	−0.243	1	−0.146	−0.144	−0.039	−0.094	−0.115	−0.137	0.062
Diabetes mellitus	−0.198	−0.146	1	−0.042	−0.051	−0.028	−0.149	−0.177	0.377
Liver diseases	−0.140	−0.144	−0.042	1	−0.035	−0.149	−0.101	−0.120	0.086
Hematological diseases	−0.072	−0.039	−0.051	−0.035	1	−0.041	−0.028	−0.033	−0.019
Dental infections	−0.202	−0.094	−0.028	−0.149	−0.041	1	−0.119	−0.142	0.056
Obesity	−0.062	−0.115	−0.149	−0.101	−0.028	−0.119	1	−0.096	−0.056
Any pathology	−0.251	−0.137	−0.177	−0.120	−0.033	−0.142	−0.096	1	−0.067
*p*-values (Pearson):									
Variables	Cardiovascular disease	Pulmonary diseases	Diabetes mellitus	Liver diseases	Hematological diseases	Dental infections	Obesity	Any pathology	Deceased
Cardiovascular disease	0	0.012	0.041	0.150	0.461	0.037	0.523	0.009	0.658
Pulmonary diseases	0.012	0	0.134	0.140	0.688	0.336	0.239	0.160	0.524
Diabetes mellitus	0.041	0.134	0	0.669	0.603	0.774	0.126	0.068	<0.0001
Liver diseases	0.150	0.140	0.669	0	0.724	0.126	0.300	0.217	0.378
Hematological diseases	0.461	0.688	0.603	0.724	0	0.677	0.778	0.737	0.845
Dental infections	0.037	0.336	0.774	0.126	0.677	0	0.221	0.145	0.570
Obesity	0.523	0.239	0.126	0.300	0.778	0.221	0	0.324	0.567
Any pathology	0.009	0.160	0.068	0.217	0.737	0.145	0.324	0	0.495

**Table 14 life-13-00535-t014:** Correlation matrix between etiological agents and therapeutic regimen.

Correlation Matrix (Pearson):				
Variables	Negative Culture	Dual Therapy	Triple Therapy	More than 3 Antibiotics
Staphylococcus aureus	−0.320	0.158	−0.104	−0.062
Streptococcus Group G	−0.234	−0.067	0.047	0.021
Streptococcus viridans	−0.209	−0.091	0.166	−0.131
Klebsiella	−0.163	−0.172	0.150	0.008
Pseudomonas aeruginosa	−0.131	0.051	0.006	−0.083
Streptococcus pyogenes	−0.106	0.003	−0.167	0.258
Fusobacterium necrophorum	−0.091	−0.096	−0.234	0.502
*p*-values (Pearson):				
Variables	Negative culture	Dual therapy	Triple therapy	More than 3 Antibiotics
Staphylococcus aureus	0.001	0.103	0.288	0.529
Streptococcus Group G	0.015	0.490	0.629	0.828
Streptococcus viridans	0.031	0.353	0.088	0.178
Klebsiella	0.094	0.077	0.124	0.932
Pseudomonas aeruginosa	0.178	0.599	0.948	0.398
Streptococcus pyogenes	0.277	0.974	0.085	0.007
Fusobacterium necrophorum	0.349	0.324	0.015	<0.0001

**Table 15 life-13-00535-t015:** Correlation between therapeutic regimen and site of suppuration.

Correlation Matrix (Pearson):			
Variables	Remedy under Treatment	Drainage with Surgery	Spontaneous Fistula
Peritonsillar space	−0.117	0.063	0.037
Submandibular space	−0.057	−0.068	0.147
Para pharyngeal space	0.163	−0.138	0.014
Buccal floor, sublingual space	0.032	0.088	−0.149
Retropharyngeal space	0.062	−0.102	0.070
Multispace	−0.028	0.092	−0.093
*p*-values (Pearson):			
Variables	Remedy under treatment	Drainage with surgery	Spontaneous Fistula
Peritonsillar space	0.229	0.518	0.703
Submandibular space	0.562	0.490	0.130
Para pharyngeal space	0.094	0.158	0.888
Buccal floor, sublingual space	0.747	0.366	0.126
Retropharyngeal space	0.524	0.298	0.476
Multispace	0.773	0.344	0.342

## Data Availability

Not applicable.

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
