# Peer review of "Principles of Treatment and Clinical-Evolutionary Peculiarities of Deep Cervical Spaces Suppurations—Clinical Study"

_life, 2023, doi:10.3390/life13020535_

Round 1

Reviewer 1 Report

I read with great interest the paper of Daniela Jicman (Stan) and colleagues, which provided a 5-year retrospective study, conducted on a group of 107 patients suffering from cervical suppurations, highlighting the particularities of deep cervical space suppurations regarding both clinical evolution and treatment. Overall, the manuscript is well-written and interesting and sheds new light on a challenging topic both from a diagnostic point of view, and from a therapeutic one. However, in the opinion of the reviewer, there is a key point that is lacking in the paper, and which is of paramount importance. The critical point is that understating the occurrence and clinical evolution of deep cervical space suppurations must be preceded by a core and deep knowledge of the anatomy of cervical fasciae. Indeed, the deep fasciae of the neck are anatomic structures with crucial clinical significance for both surgical procedures as well as in the spreading of neoplasia and infection, and suppuration, as interestingly reported in the present paper. However, the cervical fasciae have always represented a matter of debate. Indeed, in the literature, it is quite impossible to find two authors reporting the same description of the neck fascia. Thus, a complete and precise knowledge of the macroscopic anatomy of cervical fasciae and the description of the relative spatial relationships of the structures within the neck as they relate to the deep fascia by means of topographic and comparative approaches are fundamental. The studies reported in the first part of the introduction are not sufficient and they are mostly focused on the topic of “deep neck infections” rather than on the anatomy of cervical fasciae. In the opinion of the reviewer, it is important that the Author implement the introduction by discussing and citing previous anatomical studies, some key of them reported below, and clearly discuss all these points. This would add significance to the paper. Also, the discussion and conclusion should be revised considering such points.

Natale G, Condino S, Stecco A, Soldani P, Belmonte MM, Gesi M. Is the cervical fascia an anatomical proteus? Surg Radiol Anat. 2015 Nov;37(9):1119-27. doi: 10.1007/s00276-015-1480-1. Epub 2015 May 7. PMID: 25946970.

Feigl G, Hammer GP, Litz R, Kachlik D. The intercarotid or alar fascia, other cervical fascias, and their adjacent spaces - a plea for clarification of cervical fascia and spaces terminology. J Anat. 2020 Jul;237(1):197-207. doi: 10.1111/joa.13175. Epub 2020 Feb 20. PMID: 32080853; PMCID: PMC7309289.

Sutcliffe P, Lasrado S. Anatomy, Head and Neck, Deep Cervical Neck Fascia. 2022 Jul 25. In: StatPearls [Internet]. Treasure Island (FL): StatPearls Publishing; 2022 Jan–. PMID: 31082135.

Kondrup F, Gaudreault N, Venne G. The deep fascia and its role in chronic pain and pathological conditions: A review. Clin Anat. 2022 Jul;35(5):649-659. doi: 10.1002/ca.23882. Epub 2022 Apr 27. PMID: 35417568.

Author Response

Dear reviewer,
I am Dr. Jicman (Stan) Daniela, lead author of the article "Treatment principles and clinical and evolutionary characteristics of deep cervical space suppurations - clinical study", Manuscript ID life-2203430.
Thank you for your interest and recommendations for our article.
As you stated, the anatomy of the cervical region is extremely complex, so at your suggestion we have cited 3 more recommended studies.
However, the authors do not wish to debate this topic at length, but to highlight the danger of suppuration in these spaces.
In the discussion the authors compare the present study with other similar studies to emphasize the need for a diagnostic and therapeutic protocol as we have outlined in the conclusions.
We attach the revised article to your recommendation as well as the correctly numbered figures.

With kind regards,
Dr. Jicman (Stan) Daniela

Reviewer 2 Report

The article contains enormous amount of data and analyses. The file and statistics are well calculated, and the results are clearly stated. However, there is abundant amount of text, tables a graphs, which makes the article not easy to follow. 

The article is focused on Deep Neck Infections, I recommend use of this expression instead of „deep cervical space suppurations“. I recommend modifying title of the article.  The introduction is quite long and is more like for literature review, than for scientific article. I recommend shortening of introduction to one or two paragraphs.  „Starting points of suppurations“ in table 4. should be named „aetiology“.

Information given in Figures 1,2,3 are doubled in tables. 12,13,15

I recommend omitting raw 314-417. This part is suitable for group of cases study.

I do not agree with the statement  raw 563: „Currently, the management of deep cervical space suppurations is a diagnostic and therapeutic challenge, often being underestimated“ because it is nod underestimated.

The conclusion should be more precisely stated, based on result section.

Finally, I guess, this article might be of reader interest, but it requires complete re-writing, re-structuring, shortening…etc.

Author Response

Dear reviewer,
I am Dr. Jicman (Stan) Daniela, lead author of the article "Treatment principles and clinical and evolutionary features of deep cervical space suppurations - clinical study", Manuscript ID life-2203430.
Thank you for your interest and recommendations on our article.

The term suppuration implies the existence of a purulent collection, the term infection is insufficient in this case.
The authors recommend keeping the title because the patients in the study had symptoms, paraclinical examinations attesting the existence of suppurations in the deep cervical spaces.
In the introduction the authors emphasized the complex anatomy of the cervical region, as well as the deep spaces where these infectious processes can develop.In order to understand more precisely the severity of these conditions, epidemiological data, causes, clinical manifestations were mentioned....The authors consider that the information in the introduction is related to each other and recommend keeping it.
Also in table 4, the authors refer to the starting points of suppurations : odontogenic, lymphadenitis, tonsillopharyngitis, submaxillitis... considering that their identification and cure is the basis of treatment, the etiology is considered in this case : bacterial, viral, fungal...
The information presented in tables 12, 13, 15 which have been duplicated in your opinion by figures 1,2,3 , are more complex, so your recommendations are not included in the figures.
The authors recommend keeping the examples from 314-417, because all the statements in the study are demonstrated from anamnesis to multidisciplinary interventions, prognosis.
The statement in row 563, was not only found by the authors, but also by other studies (e.g. reference 17), especially patients underestimate these pathologies.
In the clinic we often found this fact.
In conclusions the authors highlight the need for an effective protocol and prevention measures through effective population scrrening on health education, oral cavity health...

With special respect,

Dr Jicman (Stan) Daniela

Reviewer 3 Report

The authors are to be commended for discussing a challenging topic. Their article provides a reminder of the difficulties treating deep neck infections, even in today's medical environment. 

Author Response

Dear reviewer,
 I am Dr. Jicman (Stan) Daniela, lead author of the article "Treatment
 principles and clinical and evolutionary features of deep cervical space suppurations – 
clinical study", Manuscript ID life-2203430.
 Thank you for your interest on our article!

Reviewer 4 Report

Congratulations on taking up an interesting topic in the manuscript regarding treatment guideline of deep cervical spaces suppurations, clinical study.  

Thank you for the opportunity to review it. Below are my comments:

The present study was undertaken to examine the deep cervical space suppurations of diagnostic and therapeutic challenges with cases that has been developed. The five year retrospective study seems very convincing and giving meaningful result to readers.

Please correct the Abstract part with correct format.

In line 28 should have left side aligned for the correspondence.

In line 32, it should have introduction section for the abstract.

Cervical deep space suppurations refer to infections that occur in the fascial planes and spaces of the head and neck. The cervical region is complex in its anatomy, with many structures enveloped by fasciae. The superficial and deep cervical fasciae are divided into sub-fasciae and these divisions create interconnected compartments that contain vital organs, blood vessels, and nerves. The superficial cervical fascia is made up of subcutaneous tissue and the platysma muscle, while the deep cervical fascia is made up of three layers: superficial, middle, and deep.

Please cite “Anatomical Proposal for Botulinum Neurotoxin Injection Targeting the Platysma Muscle for Treating Platysmal Band and Jawline Lifting: A Review” by Dr. Yi regarding platysma and neck anatomy.

These three subcomponents create cylindrical compartments that run from the base of the skull to the mediastinum. The superficial layer of the deep cervical fascia includes muscles and glands, the middle layer, called the pretracheal fascia, includes cervical organs such as the pharynx, esophagus, larynx, trachea, thyroid and parathyroid glands, and the deep layer, called the prevertebral fossa, includes the spinal column and its muscles.

The articles provide decent information with the conclusions of the clinical study suggesting that the treatment and symptomatology of deep cervical space suppurations were successfully addressed, but were affected by comorbidities such as the global health crisis and pandemic, which caused difficulties in accessing healthcare and overburdened the medical system. It is recommended that individualized treatment for deep cervical space suppurations should be approached with a multidisciplinary approach.

I would like the editor to accept the article with minor revision.

Author Response

Dear reviewer,
 I am Dr. Jicman (Stan) Daniela, lead author of the article "Treatment
 principles and clinical and evolutionary features of deep cervical space suppurations –

clinical study", Manuscript ID life-2203430.
 Thank you for your interest and recommendations on our article!

I have corrected line 28, in word format it is aligned properly.

I also corrected line 32 as you recommended.

I have included the recommended article in the references: Yi,K-H.;Lee,J-H.;Lee,K.;Hu,H-W.;Lee,H-J.;Kim,H-J.Anatomical Proposal for Botulinum Neurotoxin Injection Targeting the Platysma Muscle for Treating Platysmal Band and Jawline Lifting: A Review. Toxins.2022,14(12):868. https://doi.org/10.3390/toxins14120868

Thank you for your recommendations!

Respectfully,

Dr Jicman (Stan) Daniela

Round 2

Reviewer 1 Report

The Authors have addressed all the concerns raised by the Reviewer. I have no more academic questions.

Reviewer 2 Report

In believe, that the authors have the right to keep their own ideas. And the same belongs to rewiwer. I did not change my opinion.  The article contains a lot of information and I suggested corrections. It is on editorial board to make a decision now.  Best regards.